# PFLOTRAN-SIP: A PFLOTRAN Module for Simulating Spectral-Induced Polarization of Electrical Impedance Data

**Bulbul Ahmmed** [1,2,]*[iD]**, Maruti Kumar Mudunuru** [3][iD]**, Satish Karra** [2]**, Scott C. James** [1,4][iD]**,
Hari Viswanathan** [2] **and John A. Dunbar** [1]

[1] Department of Geosciences, Baylor University, Waco, TX 76706, USA; sc_james@baylor.edu (S.C.J.);
John_Dunbar@baylor.edu (J.A.D.)
[2] Earth and Environmental Sciences Division, Los Alamos National Laboratory, Los Alamos, NM 87545, USA;
satkarra@lanl.gov (S.K.); Viswana@lanl.gov (H.V.)
[3] Watershed & Ecosystem Science, Pacific Northwest National Laboratory, Richland, WA 99352, USA;
maruti@pnnl.gov
[4] Department of Mechanical Engineering, Baylor University, Waco, TX 76706, USA
\* Correspondence: ahmmedb@lanl.gov

**Abstract:** Spectral induced polarization (SIP) is a non-intrusive geophysical method that collects chargeability information (the ability of a material to retain charge) in the time domain or its phase shift in the frequency domain. Although SIP is a temporal method, it cannot measure the dynamics of flow and solute/species transport in the subsurface over long times (i.e., 10–100 s of years). Data collected with the SIP technique need to be coupled with fluid flow and reactive-transport models in order to capture long-term dynamics. To address this challenge, PFLOTRAN-SIP was built to couple SIP data to fluid flow and solute transport processes. Specifically, this framework couples the subsurface flow and transport simulator `PFLOTRAN` and geoelectrical simulator `E4D` without sacrificing computational performance. `PFLOTRAN` solves the coupled flow and solute-transport process models in order to estimate solute concentrations, which were used in Archie's model to compute bulk electrical conductivities at near-zero frequency. These bulk electrical conductivities were modified while using the Cole–Cole model to account for frequency dependence. Using the estimated frequency-dependent bulk conductivities, `E4D` simulated the real and complex electrical potential signals for selected frequencies for SIP. These frequency-dependent bulk conductivities contain information that is relevant to geochemical changes in the system. This study demonstrated that the `PFLOTRAN-SIP` framework is able to detect the presence of a tracer in the subsurface. SIP offers a significant benefit over ERT in the form of greater information content. It provided multiple datasets at different frequencies that better constrained the tracer distribution in the subsurface. Consequently, this framework allows for practitioners of environmental hydrogeophysics and biogeophysics to monitor the subsurface with improved resolution.

**Keywords:** parameter estimation; uncertainty analysis; forecasting; numerical modeling; Monte Carlo simulation; observation worth

## 1. Introduction

Engineered subsurface systems are dynamic due to natural and anthropogenic activities that alter porosity, permeability, fluid saturation, and geochemical properties over time [1]. Geophysical techniques, such as seismic (deep or near-surface seismic) and potential-based methods (electromagnetic, magnetic, electrical resistivity tomography (ERT), spectral induced polarization (SIP))

characterize changes in the subsurface [2–4]. Among these, ERT and SIP map the distributions of bulk electrical conductivity (i.e., the reciprocal of resistivity) due to changes in subsurface fluid flow, temperature, deformation, and reactive transport [5–9]. Because structural, topological, and geochemical properties (e.g., pore structures, fracture networks, electron donor, etc.) influence bulk electrical conductivity [2,4], ERT and SIP are applied in environmental and energy industries in order to characterize subsurface interactions. Hence, coupling ERT and/or SIP process models to flow and reactive-transport process models can enhance the interrogation of engineered subsurface systems.

ERT's data-collection component measures the electric potentials that result from an applied direct current (DC), while the data-processing component inverts these measured potentials in order to map the spatial distribution of bulk electrical conductivities [2,9,10]. ERT looks at amplitude responses, not their frequencies; therefore, it is difficult to capture multi-frequency data (typically, greater than 20 Hz). Because subsurface properties are often frequency-dependent, ERT fails to interrogate the polarization features of geologic materials, heavy metals, and induced-polarization minerals (e.g., clay minerals, hydrothermal-alteration products, pyrite, finely disseminated sulfide minerals, etc.) [2,11,12]. However, by injecting alternating currents (AC), the induced polarization (IP) method can measure "chargeability: in the time domain or "phase shift" in the frequency domain, which is the phase angle (phase lag) between the applied current and induced voltage of polarized geologic materials [13,14]. The IP method measures the energy storage capacity of certain minerals and it can be used to detect hydrocarbons [15], contaminant plumes [16–18], municipal waste, green waste (agricultural and biodegradable wastes) [19], sulfide minerals [11,20], hydrothermal products [11,20], hydrological properties [21,22], finding tree roots [23], prospecting alpine permafrost [24], investigating seawater intrusion [25], finding preferential infiltration in loess [22], and monitoring internal erosion processes [21]. IP is a single- or double-frequency method that generally fails to distinguish between a true IP response (e.g., polarized geologic materials) and noise (e.g., electromagnetic interference) [15,20]. IP signals are often low in porous geologic media and noise often masks polarization responses. Moreover, polarization responses are frequency-dependent and they reach their maxima at different frequencies. Therefore, SIP data that are collected at multiple frequencies improve subsurface imaging, even under noisy conditions.

SIP is representative of a polarization response and it cannot directly measure contaminant concentrations or chemical reactions. Coupling with a subsurface flow and reactive-transport model can tie SIP back to these processes. Furthermore, SIP is a temporal method for imaging subsurface bulk electrical conductivities. However, in practice, subsurface contaminant transport is a slow process ($\approx$1–100 years). A continuous SIP survey across a wide range of frequencies is feasible over such a long time. Consequently, SIP is performed at discrete times (snapshots) and for discrete frequencies. The acquired data are then tied back to subsurface processes through coupling to flow and reactive-transport models. The electrical conductivity from the SIP method contains information on the spatial distribution of conducting fluids and fluid chemistry. In addition, the SIP method inverts for frequency-dependent electrical conductivity that is based on measured/simulated electrical-impedance and phase-shift data, which facilitates the detection, extraction, and understanding of the evolution of hydrogeophysical and biogeophysical signatures at both the lab and field scales [26–29].

While there are numerous software for modeling geoelectrical data (e.g., IP-decay model [30], `Res2Dinv` [31–33], `Aarhusinv` [34], `BERT` [35,36], `EarthImager3D` [37], `E4D` [38], `pyGIMLi` [39], and `ZondRes3D` [40]), none capture the physics that are associated with dynamic subsurface processes. These software packages can also image frequency-dependent electrical conductivities, but they cannot capture dynamic subsurface processes. In order to overcome these problems, Johnson et al. [41] developed the massively parallel `PFLOTRAN-E4D` simulator, which couples `PFLOTRAN` [42], a subsurface flow and reactive-transport code, to `E4D`, a finite element code for simulating and inverting geoelectrical data. However, `PFLOTRAN-E4D` does not account for induced polarization. In order to capture dynamics of subsurface processes and the true sources of induced polarization, a computationally efficient framework is needed in order to couple fluid flow and solute transport with the SIP process model.



The main contribution of this study was development of a framework to include SIP in a module called `PFLOTRAN-SIP`. *This module couples subsurface flow and transport processes in* `PFLOTRAN` *with SIP capabilities in* `E4D`. This framework adds to the recently developed `PFLOTRAN-E4D` code and it leverages the `PFLOTRAN-E4D` massively parallel capabilities. We demonstrated `PFLOTRAN-SIP` capabilities with a representative tracer-transport process in a medium with polarization properties. The primary focus of this modeling exercise was to illustrate the capabilities of the `PFLOTRAN-SIP` framework not to validate the proposed model.

The paper is organized, as follows: Section 1 discusses the limitations of ERT along with the advantages of the SIP method. The importance of coupling between fluid flow, reactive-transport, and SIP process models is discussed. A discussion on the state-of-the-art simulators is also provided. Section 2 introduces the `PFLOTRAN-SIP` framework. The methodology for coupling `PFLOTRAN` and `E4D` simulators is described. Process models that are related to SIP, fluid flow, and solute transport in `PFLOTRAN` and `E4D` simulators are presented. Additionally, this section discusses the mesh interpolation that transfers state variables between the `PFLOTRAN` and `E4D` meshes. Section 3 describes a reservoir-scale model setup in order to perform high-fidelity numerical simulations. It describes the model domain, related meshes, initial conditions, boundary conditions, and various input parameters that are related to the `PFLOTRAN-SIP` framework. Additionally, this section provides details on the inversion procedure for electrical conductivity at different frequencies. This exercise was performed with the intention of comparing the simulated conductivity/tracer distribution from the `PFLOTRAN-SIP` framework and inverted electrical conductivity from SIP inversion. Section 4 explains simulated electrical potentials for a measurement and then compares the true/simulated and estimated conductivities. It shows that the estimated frequency-dependent conductivities from the SIP inversion were consistent with the tracer concentration/simulated conductivity from the `PFLOTRAN-SIP` framework. Section 4 also provides a discussion on the numerical results, limitations of the proposed framework, and how geoscientists can use the `PFLOTRAN-SIP` framework for their applications. Finally, conclusions are drawn in Section 5.

## 2. PFLOTRAN-SIP: Process Models and Coupling Framework

The `PFLOTRAN-SIP` framework couples flow and reactive-transport process models in `PFLOTRAN` [42–45] with the SIP process model in `E4D` [38,46,47] to characterize fluid-driven electrical impedance signatures across multiple frequencies. At each time-step, the simulation outputs from `PFLOTRAN` (fluid saturation, tracer concentration, etc.) were supplied to Archie's Law [48] to calculate fluid-dependent bulk electrical conductivities for `E4D` simulations. These estimated bulk electrical conductivities were decomposed into real and imaginary components for each frequency while using the Cole–Cole model [49,50], which is an empirical description of frequency-dependent behavior of bulk electrical conductivities. These processes were repeated until the entire transient simulation was completed.

### 2.1. E4D Process Model

`E4D` is an open-source, massively parallel, finite-element code for simulating and inverting three-dimensional time-lapsed ERT and SIP data [38,46,47,51]. The process models in `E4D` for ERT and SIP assume that the displacement currents are negligible and current density can be described by Ohm's constitutive model [51]. These assumptions result in a Poisson equation relating induced current to the electric potential field:

$$- \mathrm{div}\left[\sigma\left(\mathbf{x}\right)\mathrm{grad}\left[\Phi_\sigma\left(\mathbf{x}\right)\right]\right] = \mathcal{I}\delta\left(\mathbf{x} - \mathbf{x}_0\right), \tag{1}$$

where $\sigma\,[\mathrm{S\,m^{-1}}]$ is the effective electrical conductivity, $\mathcal{I}\,[\mathrm{A}]$ is the injected current, and $\Phi_\sigma(\mathbf{x})\,[\mathrm{V}]$ the electrical potential all at position-vector $\mathbf{x}\,[\mathrm{m}]$, $\delta\left(\cdot\right)$ is the Dirac delta function, div is the divergence of a vector field, while grad is the gradient of a scalar field [52,53].

Equation (1) models the DC effect, which is required in ERT forward/inverse modeling; however, it does not account for induced polarization under AC. IP under AC results in a secondary potential that needs to be accounted for in the SIP forward/inverse modeling. This requires the modification of Equation (1) in order to solve for the total electrical potential field under IP effects:

$$- \operatorname{div} \left[ (1 - \eta(\mathbf{x})) \, \sigma(\mathbf{x}) \operatorname{grad} \left[ \Phi_\eta(\mathbf{x}) \right] \right] = \mathcal{I} \delta(\mathbf{x} - \mathbf{x}_0), \tag{2}$$

where $\Phi_\eta$ [V] is the total electrical potential field, which includes IP effects from a polarized material with chargeability distribution $\mathbf{\jmath}(\mathbf{r})$ [mrad] [54]. The secondary potential that results from the IP effect is [55]:

$$\Phi_\mathrm{s} = \Phi_\eta - \Phi_\sigma, \tag{3}$$

and the apparent chargeability is [54]:

$$\eta_\mathrm{a} = \frac{\Phi_\eta - \Phi_\sigma}{\Phi_\eta}. \tag{4}$$

The secondary potential $\Phi_\mathrm{s}$ and apparent chargeability $\eta_\mathrm{a}$ are weakly nonlinear, which result from Equations (1) and (2). These potentials $\Phi_\eta$, $\Phi_\sigma$, and $\Phi_\mathrm{s}$ are time-domain signatures of induced polarization. Equation (3) is in the time domain and it is transformed into the frequency domain:

$$- \operatorname{div} \left[ \sigma^*(\mathbf{x}, \omega) \operatorname{grad} \left[ \Phi^*(\mathbf{x}) \right] \right] = \mathcal{I} \delta(\mathbf{x} - \mathbf{x}_0), \tag{5}$$

where $\omega$ [Hz] is the frequency. $\sigma^*(\mathbf{x}, \omega)$ [$\mathrm{S\,m^{-1}}$] and $\Phi^*(\mathbf{x})$ [V] are the frequency-dependent electrical conductivities and electrical potential, respectively. $\Phi^*(\mathbf{x})$ includes the real and imaginary electrical potentials that correspond to induced polarization. Zero potential is enforced on boundaries of the domain ([51], Section 3) in order to solve Equation (5).

E4D simulates four-electrode configurations (e.g., Wenner and dipole–dipole arrays) [38]. Current is injected from source to sink electrodes, while measurements are recorded between the other two electrodes [3,8,38]. For ERT, the measured response is the potential difference (voltage) between the two electrodes, while SIP also includes the phase shift (radians). Based on the user-defined survey design, E4D can simulate up to thousands of ERT/SIP measurements to compute electrical potential distributions. Because the governing equations are linear in $\Phi_\sigma$ and $\Phi_\eta$, E4D solves Equation (5) by superimposing pole solutions with different current sources that makes ERT or SIP forward modeling highly scalable [38,51].

E4D solves the ERT and SIP process models in the frequency domain while using a low-order finite element method (FEM). The output of the FEM solution for the ERT process model is electrical potential throughout the domain, which is real valued and frequency independent. Because the SIP process model is frequency dependent, the corresponding output of the FEM solution has both real and imaginary components of electrical potential. The complex-valued electrical potential (or, equivalently, the phase-shift distribution in the model domain) provides new information on IP in the subsurface, which is not capturable by ERT.

E4D uses the standard Galerkin weak formulation [56] on an unstructured, low-order, tetrahedral, finite element mesh [57], and it iteratively computes the total electrical potential field due to IP effects ([51], Section 3). Equations for computing the real and imaginary components of the complex-valued electrical potential are decoupled, and the finite-element analysis is performed in the real-number domain. First, E4D solves for the real component without considering the IP effects. Second, the current source for the imaginary component is computed from the real component. Third, the imaginary component of the total electrical potential is calculated based on this computed current source. Fourth, the secondary current source arising from the imaginary component is computed. This secondary current source considers the IP effects. Later, the real component is calculated based on this secondary current source. These steps are repeated until a convergence criterion is satisfied.

## 2.2. PFLOTRAN Process Models

PFLOTRAN solves a system of nonlinear partial differential equations describing multiphase, multicomponent, reactive flow, and transport while using the finite-volume method (FVM) [42,43,58]. In this paper, we only consider single-phase fluid flow and solute transport when predicting the spatio-temporal distribution of solute concentrations. Mass conservation for single-phase, variably saturated flow is:

$$\frac{\partial \phi s \rho}{\partial t} + \mathrm{div}\,[\rho \mathbf{q}] = Q_{\mathrm{w}}, \tag{6}$$

where $\rho$ [kg m$^{-3}$] is the fluid density, $\phi$ [–] is the porosity, $s$ [–] is the saturation, $t$ [s] is time, $\mathbf{q}$ [m s$^{-1}$] is the Darcy flux, and $Q_{\mathrm{w}}$ [kg m$^{-3}$ s$^{-1}$] is the volumetric source/sink term. Darcy flux is:

$$\mathbf{q} = -\frac{\kappa \kappa_{\mathrm{r}}\,(s)}{\mu} \mathrm{grad}\,[p - \rho g z], \tag{7}$$

where $\kappa$ [m$^2$] is the intrinsic permeability, $\kappa_{\mathrm{r}}$ [–] is the relative permeability, $\mu$ [Pa s] is dynamic viscosity, $p$ [Pa] is pressure, $g$ [m s$^{-2}$] is gravity, and $z$ [m] is the vertical component of $\mathbf{x}$. The source/sink term is:

$$Q_{\mathrm{w}} = \frac{q_{\mathrm{M}}}{W_{\mathrm{w}}} \delta\,(\mathbf{x} - \mathbf{x}_Q), \tag{8}$$

where $q_{\mathrm{M}}$ [kg m$^{-3}$] is the mass flow rate, $W_{\mathrm{w}}$ [kg kmol$^{-1}$] is the formula weight of water, and $\mathbf{x}_Q$ [m] denotes the location of the source/sink. The governing equation for tracer transport is:

$$\frac{\partial \phi c}{\partial t} + \mathrm{div}\,[c\mathbf{q} - \phi s \tau D\,\mathrm{grad}\,[c]] = Q_c, \tag{9}$$

where $c$ [molality] is the solute concentration, $D$ [m$^2$ s$^{-1}$] is the diffusion/dispersion coefficient, $\tau$ [–] is tortuosity, and $Q_c$ [molality s$^{-1}$] is the solute source/sink term. Dirichlet, Neumann, or Robin boundary conditions are specified when solving Equations (6)–(9).

Coupled governing Equations (6)–(9) are solved with a two-point flux FVM in space and a fully implicit backward Euler method in time while using a Newton-–Krylov solver [43,59]. PFLOTRAN consists of master process $\mathcal{A}$, child process $\mathcal{B}$, and peer process $\mathcal{C}$ (see, Figure 2 of Johnson et al., (2017) [41]). Here, the flow model is master process $\mathcal{A}$, while $\mathcal{B}$ and $\mathcal{C}$ are the solute transport and E4D/SIP models, respectively. The time step for the flow model may be different from the solute-transport model. The transfer of information between $\mathcal{A}$ (e.g., flow) and $\mathcal{B}$ (e.g., solute transport) takes place before and after each of $\mathcal{A}$'s time steps. The synchronization of $\mathcal{A}$ and $\mathcal{C}$ (e.g., ERT or SIP) occurs at specified times. Execution starts with the master-process model $\mathcal{A}$, which can take as many adaptive time steps as needed to reach the synchronization point. $\mathcal{B}$ and $\mathcal{C}$ proceed according to their time steps ($\leq \mathcal{A}$'s) in order to reach the synchronization point. When $\mathcal{A}$, $\mathcal{B}$, and $\mathcal{C}$ all reach the synchronization point, variables and parameters (e.g., saturation, solute concentration, porosity, etc.) are updated between $\mathcal{A}$ and $\mathcal{C}$.

## 2.3. PFLOTRAN-SIP Coupling

Coupling involves six steps: (1) PFLOTRAN's flow model calculates fluid pressure, saturation, and velocity; (2) while using those simulated outputs, the transport model calculates solute concentrations; (3) solute concentrations in each PFLOTRAN mesh cell are used to calculate DC electrical conductivities for ERT based on Archie's law; (4) the Cole–Cole model is used to calculate frequency-dependent electrical conductivities; (5) real and imaginary electrical conductivities are interpolated onto the E4D mesh; and, (6) the SIP model solves the forward problem to calculate electrical impedance and phase shifts.

PFLOTRAN and E4D use Message Passing Interface calls for inter-process communication. Based on user specification, PFLOTRAN divides the computing resources between PFLOTRAN and E4D at the initial

step. PFLOTRAN and E4D read their corresponding input files and complete pre-simulation steps. These include setup of the flow model, the solute transport model, the SIP model, and the mesh interpolation matrix. Mesh interpolation is needed for two reasons: (1) the meshes of PFLOTRAN and E4D are different and (2) the solution procedure of PFLOTRAN is based on the FVM, while E4D's solution procedure is based on the FEM. Consequently, the state variables (e.g., solute concentration, fluid saturation) computed at the cell center by PFLOTRAN need to be accurately transferred from the PFLOTRAN mesh to the E4D mesh in order to calculate electrical conductivities. Section 2.5 describes the generation of the mesh interpolation matrix. Algorithm 1 and Figure 1 summarize the coupling of PFLOTRAN and SIP models.

---

**Algorithm 1** Overview of the proposed PFLOTRAN-SIP framework for simulating electrical impedance data.

---

1: INPUT: Initial and boundary conditions for fluid flow and solute transport models in PFLOTRAN, fluid density, porosity, saturation, volumetric source/sink with its location, intrinsic and relative permeabilities, dynamic viscosity, mass flow rate, diffusion/dispersion coefficients, tortuosity, solute source/sink with its location, Archie's and Cole-Cole model parameters, total simulation time, time-step for PFLOTRAN, interrogation frequencies, electrode locations and measurement configuration, number of processors for PFLOTRAN and E4D, and meshes for PFLOTRAN and E4D.

2: Solve Equations (6)–(8) for fluid pressure, fluid saturation, and fluid velocity.

3: Solve Equation (9) to calculate the spatio-temporal distribution of solute concentration.

4: Transfer solute concentration from PFLOTRAN to the E4D master processor to perform SIP simulations at specific times.

5: Receive numerical model setup information from PFLOTRAN input files to perform mesh interpolation for SIP simulations.

6: Broadcast run information and distribute mesh assignments to E4D slave processors.

7: Calculate the mesh interpolation matrix to interpolate PFLOTRAN simulation outputs (e.g., solute concentrations) onto the E4D mesh for SIP simulations.

8: Calculate electrical conductivities using Archie's model Equation (10).

9: Calculate frequency-dependent electrical conductivities using the Cole-Cole model Equation (11).

10: Pass real and imaginary conductivities calculated at different frequencies to the E4D master processor to perform SIP simulations.

11: Broadcast real and imaginary conductivities to E4D slave processors to compute pole solutions for electrode configurations.

12: Solve Equation (5) to compute complex electrical potential at different frequencies and solute concentrations at specified times.

---

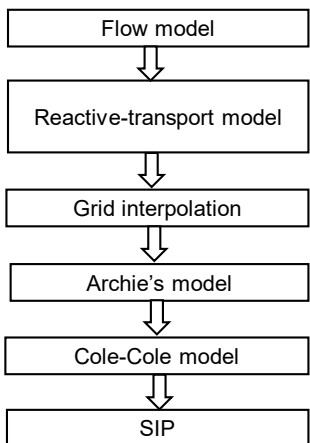

**Figure 1. Coupling PFLOTRAN and spectral induced polarization (SIP) Process Models:** Steps involved in coupling fluid flow, solute transport, and SIP process models in the PFLOTRAN-SIP framework. Details of the inputs, data, process models, and outputs are described in Algorithm 1.

### 2.4. Petrophysical Transformation

A mathematical relationship linking fluid flow state variables and bulk electrical conductivities is required in order to simulate SIP signals during fluid flow and solute transport. Archie's Law [48,60,61] is a petrophysical transformation relating state variables simulated by `PFLOTRAN` in order to bulk electrical conductivities:

$$\sigma_b\left(\mathbf{x}\right) = \frac{1}{\tau_f}\phi^\alpha s_f^\beta \sigma_f, \tag{10}$$

where $\tau_f$ [–] is the tortuosity factor (path length of current), $\sigma_b(\mathbf{x})$ [S m$^{-1}$] is the bulk electrical conductivity at near-zero frequency ($\omega \sim 0$), $\alpha$ [–] is the cementation exponent (1.8 to 2.0 for sandstone), $s_f$ [–] is the solute concentration that is simulated by `PFLOTRAN`, $\beta$ [–] is the saturation exponent (close to 2.0), and $\sigma_f$ [S m$^{-1}$] is the fluid electrical conductivity.

In order to account for frequency dependence, Equation (10) was modified using the Cole–Cole model [49,50,62–64]:

$$\sigma^*\left(\mathbf{x}, \omega\right) = \sigma_b\left(\mathbf{x}\right)\left\{1 + \eta_a\left[\frac{(i\omega t_r)^\gamma}{1 + (1 - \eta_a)(i\omega t_r)^\gamma}\right]\right\}, \tag{11}$$

where $i^2 = -1$, $\gamma$ [–] is a shape parameter and $t_r$ [s] is the characteristic relaxation time constant (time for the imaginary electrical component to reach equilibrium after perturbation) that is related to characteristic pore or grain size.

### 2.5. Mesh Interpolation

Once the frequency-dependent real and imaginary components of bulk electrical conductivities were calculated on the `PFLOTRAN` mesh, they were interpolated onto the `E4D` mesh. The conductivity at any intermediate point in a `PFLOTRAN` mesh cell was approximated while using tri-linear interpolation. Tri-linear interpolation is a multivariate interpolation function on a three-dimensional regular grid. It linearly approximates the value of a function at an intermediate point $(x, y, z)$ within the local rectangular prism, using function data on the lattice points. Here, approximated values were computed while using values at the `PFLOTRAN` cell centers surrounding the point at `E4D` grid [41].

## 3. Methodology

### 3.1. PFLOTRAN Model Setup

A simple example model was developed in order to demonstrate `PFLOTRAN`– `SIP`. Similar to the Hanford Site, Richland, Washington [41], a uniform pressure gradient drove flow in the positive $x$ direction. The system was intended to be representative of sandstone with an intermittent shale layer. This synthetic problem included contaminant transport with the intention to support remediation by providing insight into the evolution of the tracer distribution. The domain was $500 \times 500 \times 500$ m$^3$ and it consisted of three layers, as shown in Figure 2. The upper layer was $500 \times 500 \times 350$ m$^3$ and extended from $z = 0$ to $-350$ m as a highly conductive material with $\kappa = 7.38 \times 10^{-13}$ m$^2$ (see, Table 1). The fluid was assumed to be water while rock properties (e.g., $\kappa$, $\phi$, $D$, etc.) were assumed to be sandstone. The middle layer was less permeable ($\kappa = 1.05 \times 10^{-22}$ m$^2$) with size $500 \times 500 \times 50$ m$^3$ extending from $z = -350$ to $-400$ m. This $\kappa$ is representative of shale or granite. However, the low-permeability layer included a small-volume, sandstone ($\kappa = 7.38 \times 10^{-13}$ m$^2$) material between $x = 300$ and $350$ m, $y = 0$ and $500$ m, and $z = -400$ and $-450$ m. The bottom layer was also sandstone ($\kappa = 7.38 \times 10^{-13}$ m$^2$), with dimensions of $500 \times 500 \times 100$ m$^3$.

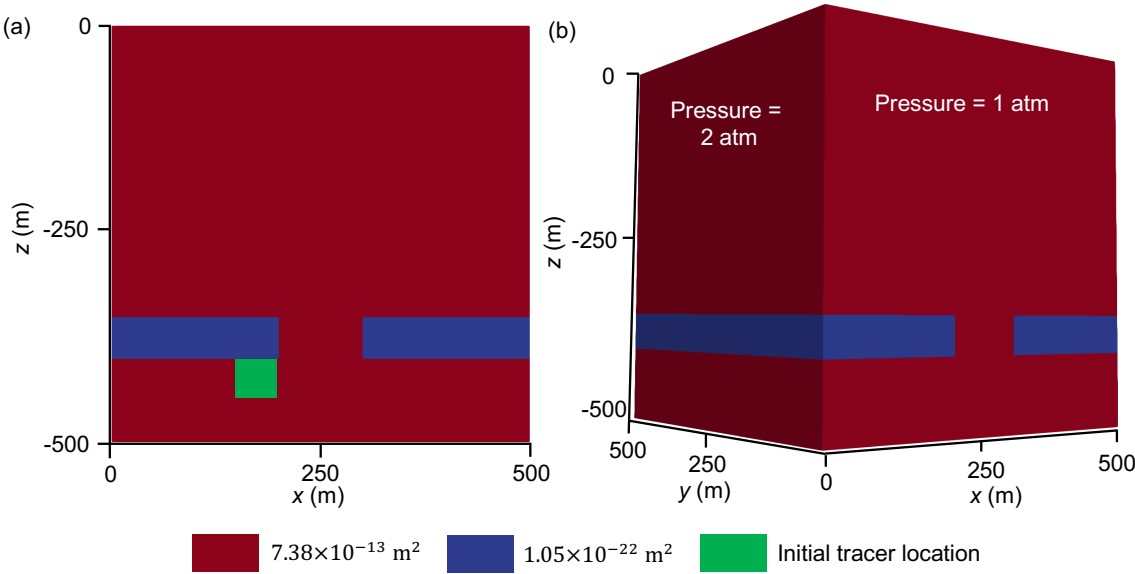

**Figure 2. PFLOTRAN model domain:** Schematics of (**a**) permeability distribution and (**b**) pressure boundary conditions.

A solute (conservative tracer) at $10\,\text{mol/kg}$ was placed below the low permeable zone, as shown in Figure 2 as the green $50 \times 500 \times 50\,\text{m}^3$ block. The initial and boundary conditions for the model included: pressure of 1 atm at the top with a hydrostatic pressure gradient from top to bottom. The left face ($x = 0$) was assigned a hydrostatic pressure of 2 atm in order to drive flow from left to right. For solute transport, the boundary conditions were zero-concentration Dirichlet inflow at the left face and zero diffusive gradient outflow at the right face that allowed only advective outflow. The remaining faces were specified as zero-solute flux boundaries.

For low- and high-$\kappa$ zones, $\tau = 1$, while $\phi$ were 0.3 and 0.25, respectively. The solute diffusivity was $10\,\text{m}^2\,\text{s}^{-1}$. The Newton solver (20-iteration maximum) was applied for flow and solute transport. For the flow solver, relative and absolute tolerances [–] were $10^{-50}$ with a relative update tolerance of $10^{-60}$, while, for solute transport solver, the relative and absolute tolerances were $10^{-4}$ with a relative update tolerance of $10^{-60}$. The simulation was run for one year with an initial time step of $10^{-8}$ years, which was allowed to accelerate by a factor of 8.

**Table 1.** PFLOTRAN and SIP parameters used in the model and corresponding values.

| PFLOTRAN Parameters | Values | SIP Parameters | Values |
|---|---|---|---|
| $\kappa$ (top & bottom layers) | $7.38 \times 10^{-13}\,\text{m}^2$ | Injected current | 1 A |
| $\kappa$ (middle layer) | $1.05 \times 10^{-22}\,\text{m}^2$ | $\sigma$ | heterogeneous values $\text{Sm}^{-1}$ |
| Initial solute concentration | $10\,\text{mol\,kg}^{-1}$ | $\alpha$ | 0.564 |
| $\phi$ (top and bottom layers) | 0.3% | $\beta$ | 0.576 |
| $\phi$ (middle layer) | 0.25% | $t_r$ | 0.061 |
| Diffusivity | $10\,\text{m}^2\,\text{s}^{-1}$ | $\omega$ | 0.1, 1, 10, 100, and 1000 Hz |

*3.2. SIP Model Setup*

Although the domain dimensions for SIP simulations were identical to the PFLOTRAN simulation, there was only a single layer. The corresponding E4D mesh for the simulation had 86,780 nodes and 609,562 tetrahedral elements. In order to avoid zero potentials effects on the SIP model, zero potentials were enforced on the external boundaries, which were 9500 m away from each lateral boundary, except for the top, which corresponded to the ground surface in both models. This extension of the SIP model domain aided the SIP simulation [38]. A total of 80 point electrodes were placed in the

domain, all located at $z = -425\,$m arranged in five lines along the $x$-axis, with each line comprising 16 electrodes. The electrode coordinates started at $(40, 50, -425)$ and ended at $(460, 450, -425)$ with a $100\,$m separations between lines see, Figure 3. Although, in practice, it is much easier to place electrodes on the surface, in this simulation they were placed in the region of interest (i.e., at depth) to provide more accurate data that facilitated a better inversion of subsurface properties and processes. When compared to surface-lain electrodes, electrodes buried at depth are less impacted by noise (e.g., due to anthropogenic activities). Electrode measurement configurations included a combination of Wenner and dipole–dipole arrays.

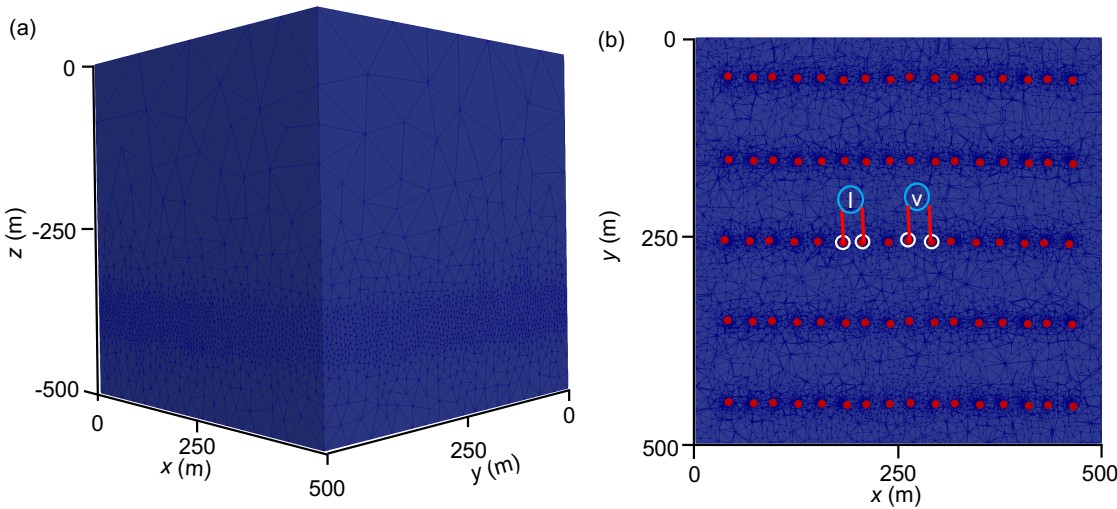

**Figure 3. SIP model:** (**a**) The three-dimensional (3D) SIP model domain where (**b**) red dots represent electrodes on the $xy$-plane. White circles represent electrode configuration of 80th out of 1062 simulated electrical impedance where I and V represent current and potential electrodes, respectively.

A current of 1A was injected and received at a pair of electrodes, and the potential difference was measured at another pair of electrodes. There are various advantages of injecting and receiving the current through a pair of electrodes. For example, such a measurement system can eliminate any inaccuracies that are caused by the injecting circuit impedance (the contact impedance between the probe and the medium, which can be high). While using the prescribed measurement configuration, a total of 1062 simulated measurements were collected in order to capture electrical impedance and phase shift.

The electrical conductivity of the fluid at $\omega = 0\,$Hz was $2 \times 10^{-3}\,$S m$^{-1}$. Parameters $\alpha$, $\beta$, and $t_{\mathrm{r}}$ were 0.564, 0.576, and 0.061 s, respectively, all being representative of sandstone [65] (see, Table 1). SIP analysis was performed for five different frequencies: 0.1, 1, 10, 100, and 1000 Hz. Forward model simulations were performed while using 61 processors, where 20 processors were assigned for PFLOTRAN and 41 for E4D. Out of those 41 processors, 40 performed SIP simulations for different measurement configurations, and the remaining processors gathered the simulated data.

## 3.3. SIP Inversion of Electrical Conductivity

For verification, E4D's inversion module was used in order to estimate frequency-dependent electrical conductivity that is based on the simulated electrical impedance and phase-shift data. This estimated conductivity was compared with the simulated conductivity that was generated by the PFLOTRAN-SIP framework. The employed inversion process was blind (i.e., we did not provide prior constraints on the conductivity). This can be improved by providing detailed conductivity information to E4D's inversion module. The SIP inversion employs an unstructured mesh, which consisted of 51,124 nodes with 316,183 mesh elements. Low-order mesh elements were generated to make the inversion process simple and computationally efficient, because high-order

mesh elements did not improve the resolution of the electrical conductivity [47]. However, the meshes were refined around electrodes, where the volume of each mesh was in the order of $cm^3$. Simulated measurements (electrical impedance) by `PFLOTRAN-SIP` were the data supplied to the inversion process as observations.

E4D was inverted by minimizing the following objective function:

$$\Phi = \Phi_d \left[ W_d \left( \Phi_{obs} - \Phi_{pred} \right) \right] + \zeta \Phi_m \left[ W_m \left( \sigma_{est} - \sigma_{ref} \right) \right], \tag{12}$$

where $\Phi_d$ is a scalar operator that quantifies the misfit between observed and simulated data (e.g., electrical impedance and phase shift) based on the user-specified norm (e.g., Euclidean norm), $\Phi_m$ is another operator that provides a scalar measure of the difference between the frequency-dependent electrical conductivity distribution, $\sigma_{est}$ [S m$^{-1}$], and constraints placed upon the structure of $\sigma_{ref}$ [S m$^{-1}$], $\zeta$ is the regularization parameter, $W_d$ is the data-weighting matrix, and $W_m$ is the model-weighting matrix. $\sigma_{est}$ and $\sigma_{ref}$ are the estimated and reference frequency-dependent electrical conductivities. The user specified bounds on the frequency-dependent conductivity in each mesh cell were 0.000 01 and 1.0. The $\Phi_{obs}$ and $\Phi_{pred}$ were the observed and simulated data, respectively. Equation (12) is solved while using the iteratively reweighted least square method [66]. Further details on the parallel inverse modeling algorithm and its implementation in E4D are available [38].

The $\zeta$ value was 100 at the beginning of the inversion and it decreased as the nonlinear iteration progressed. Before $\zeta$ was reduced, the minimum fractional decrease in the objective function, $\Phi$, between iterations had to be <0.25 whereafter $\zeta$ was reduced to 0.5. The convergence of the SIP inversion procedure was based on the $\chi^2$ value of the current iteration after data culling, being computed as:

$$\chi^2 = \frac{\Phi_d}{n_d - n_c}, \tag{13}$$

where the data residual is the difference between observed and estimated values divided by the standard deviation for that measurement. $n_d$ is the total number of survey measurements and $n_c$ is the number of measurements that were selected from the total number of measurements during the current iteration.

## 4. Results & Discussion

The one-year `PFLOTRAN-SIP` model simulations were completed in two minutes. The computation was performed on 61 Intel® Xeon® CPU E5-2695 V4 @ 2.1 GHz processors. Figure 4 shows the tracer concentrations at the end of the simulation. In one year, the pressure gradient drove tracers about 100 m from its initial location in the $x$-direction and also moved it upward about 20 m.

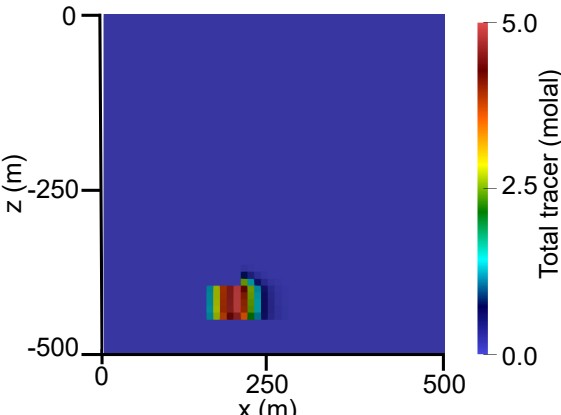

**Figure 4. PFLOTRAN Simulation:** Spatial distribution of tracer concentrations after one year.

The SIP module in `PFLOTRAN-E4D` simulated real and imaginary electrical impedance at 0.1, 1, 10, 100, and 1000 Hz. Because of minimal differences between 1 and 10 Hz, only results for 0.1, 10, 100, and 1000 Hz are discussed. This indicated that some frequencies may be redundant, because they yield similar impedance. Sensitivity analyses can be performed in order to identify redundant frequencies; however, this was beyond the scope of this paper. Figure 5 shows the real and imaginary potentials due to changes in tracer concentration for the various frequencies. Additionally, this figure provides information on the change in electrical potential at different frequencies for a single measurement configuration, indicating the maximum tracer concentration. The 80th out of 1062 electrical impedance measurements (see, Figure 3b) was selected where the tracer concentrations were the most evident. The response clearly shows the polarization feature of the tracer. The gradient of the real electrical potential was high near $x = 300$ m (top row of Figure 5), where tracer concentrations were maximum. From Figure 5, it is evident that the real potential response for 0.1 Hz was different from the responses at 10, 100, and 1000 Hz. The root-mean-square error (RMSE) between these responses was approximately 15% of the maximum real potential value, indicating that frequency has an impact on the real potential distribution.

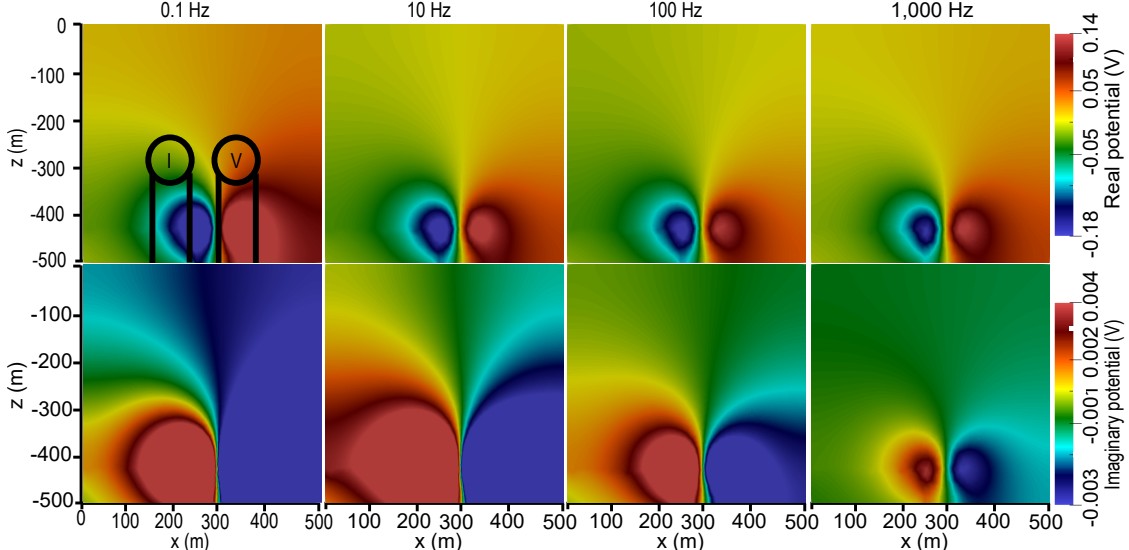

**Figure 5.** Slices of simulated real (**top**) and imaginary (**bottom**) components of complex electrical potentials/impedances at $y = 250$ m for a single measurement after one year. The measurement location is in top left-corner plot of this figure and in Figure 3b.

The bottom row of Figure 5 shows the imaginary component of complex electrical potential responses where the polarity was switched (colors interchanged). Unlike the real electrical potential, each imaginary electrical potential was notably different, indicating its frequency dependence. The corresponding RMSE between responses was ∼85% of the maximum imaginary potential value. Such a high variation was expected, as the imaginary electrical potential depends on frequency, chargeability, and relaxation time, although the last two were constant in this study. Because the response of the imaginary potential was clearly visible in the simulation, this indicated that the `PFLOTRAN-SIP` framework can effectively simulate polarized geologic materials.

Figure 6 shows the simulated and estimated frequency-dependent electrical conductivities while using the `PFLOTRAN-SIP` framework with the SIP inversion module in E4D. The true (`PFLOTRAN` simulated) and estimated (inversion of survey data) real electrical conductivities are plotted in Figure 6a–d and Figure 6e–h, respectively. SIP inversion was performed while using the simulated electrical impedance, and the phase-shift data obtained from `PFLOTRAN-SIP` model runs after one year. Inversion converged after 48 iterations when $\chi^2$ reached 60. The computational time that was required to perform SIP inversion was approximately two hours on 41 Intel® Xeon® CPU E5-2695 V4

processors running at 2.1 GHz. Estimated electrical conductivities showed high contrast around the high tracer distribution/simulated conductivities, although they were more diffuse than the true (simulated) distribution (Figure 6a–h). ERT provided data for Figure 6e, but SIP provided data for Figure 6e–l. Although not all SIP data were informative, some were useful. For example, the estimated conductivities at 1000 Hz were more accurate than frequencies <1000 Hz with the same inversion constraints. Later, real conductivity values were used in Equation (11) in order to provide initial guesses for imaginary conductivities for SIP inversion. Figure 6i–l shows the estimated imaginary electrical conductivity distributions. Similar to estimated real conductivities, imaginary conductivities that were computed from SIP inversion were also diffuse. The inversion process could be improved by providing prior information and structural constraints on electrical conductivities. However, both estimated conductivities were generally consistent with the tracer distribution, which showed that the SIP inversion module can simulate electrical impedance and phase-shift data. To summarize, SIP provides a major benefit over ERT in the form of greater information content. This is because an SIP survey yields multiple datasets at different frequencies that help to overcome false positives (i.e., indication of a tracer where none is present). For example, from Figure 6, it is evident that the SIP inversion analyses at different frequencies revealed the same tracer region (not a false positive). With an ERT survey, it may be difficult to identify a false positive from a true positive, because ERT only generates a single dataset.

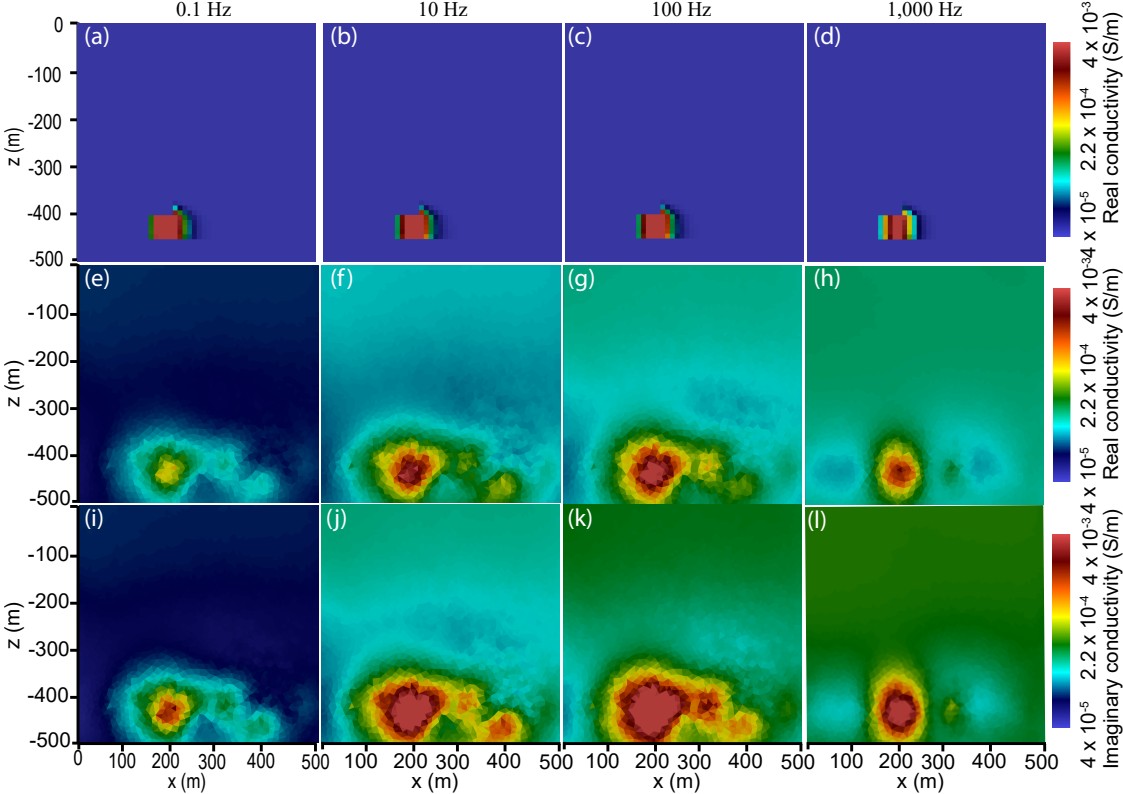

**Figure 6.** Simulated and estimated frequency-dependent electrical conductivities at $y = 250$ m after one year. (**a–d**) True-electrical conductivities from the `PFLOTRAN-SIP` framework, (**e–h**) estimated bulk-real conductivities from SIP inversion, and (**i–l**) imaginary components of estimated bulk complex electrical conductivities from SIP inversion.

Figure 7a–c show the simulated outputs of tracer concentrations, real potentials, and imaginary potentials for the 80-electrode measurement configuration at frequencies of 0.1, 10, 100, and 1000 Hz. The location of maximum tracer concentration was around $x = 300$ m (Figure 7a). The locations of current and potential measurement electrodes were at ($x = 208$, 236, 264, and 292 m, $y = 250$ m,

and $z = -420\,\mathrm{m}$) (Figure 3b). Note that the electrodes were not placed at the location of maximum concentration, but they were placed 50 m right of maximum concentration in a line in the subsurface. Nevertheless, the measured potentials provided meaningful information on the bounds of the tracer distribution as well as revealing the significance of higher frequencies that were obtained from a combination of electrical impedance and phase shift.

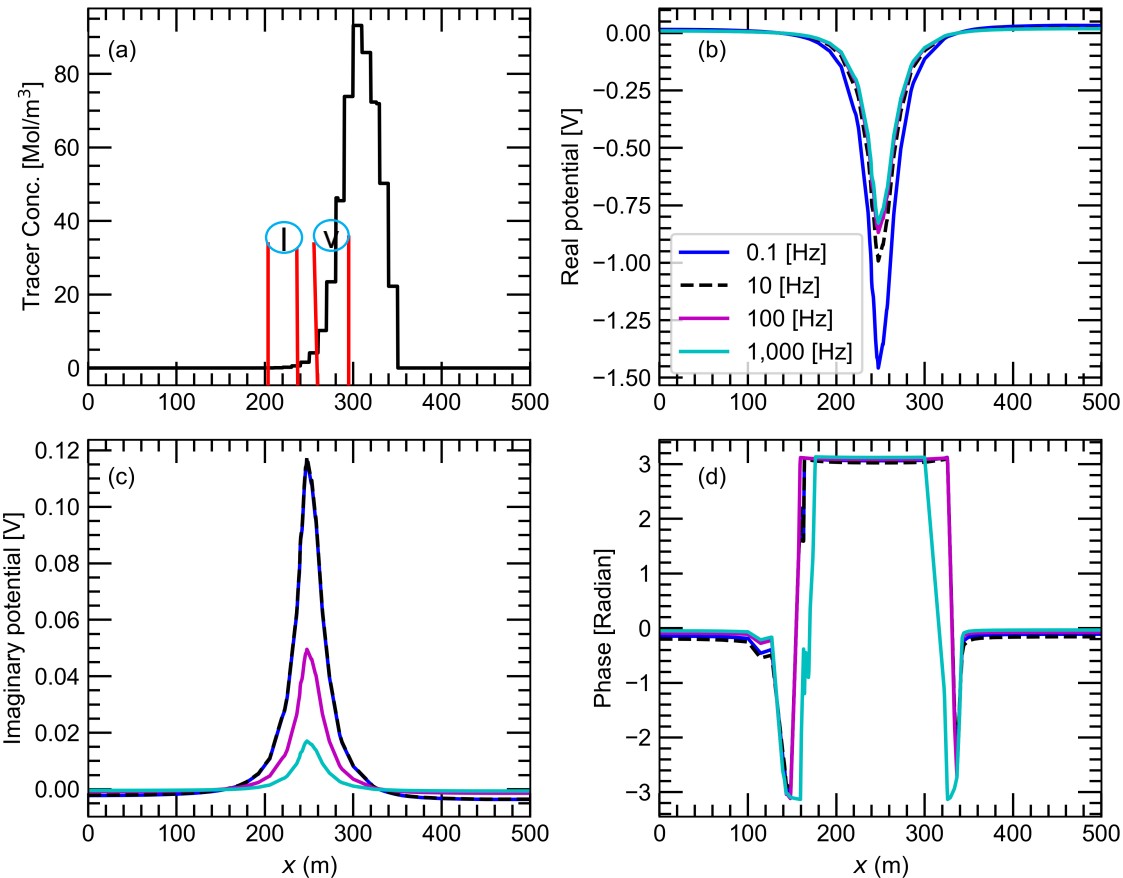

**Figure 7.** Distribution of (**a**) tracer concentration and (**b**) real potential, (**c**) imaginary component of complex potential, and (**d**) phase shift along the *x*-axis at $y = 250\,\mathrm{m}$ and $z = -425\,\mathrm{m}$.

For this study, only tracer concentration impacted the real and imaginary components of complex conductivities revealed through the Cole-Cole model because $\alpha$, $\beta$, and $t_r$ were held constant in order to investigate the effect of tracer concentration over different frequencies. Figure 8a,b show how the Cole–Cole model increased real conductivities and decreased the imaginary component of complex conductivities over different frequencies. Figure 7b,c shows that the absolute real potential and imaginary potential decreased as the frequency increased. E4D first solved the real potential, $\Phi_r$, as noted in Section 2.1 and in by [51], for SIP simulations. That is, $-\mathrm{div}\left[\sigma_r \mathrm{grad}\left[\Phi_r\right]\right] = I$, where $\sigma_r$ is the real component of $\sigma^*(\mathbf{x}, \omega)$ and $\Phi_r$ is inversely proportional to $\sigma_r$. Additionally, $\sigma_r$ increased as $\omega$ increased; hence, the absolute value of the real potential distribution (as shown in Figure 7b) decreased as $\omega$ increased. After $\sigma_r$ was evaluated, E4D computed the complex potential by solving $\mathrm{div}\left[\sigma_r\mathrm{grad}\left[\Phi_c\right]\right] = -\mathrm{div}\left[\sigma_c\mathrm{grad}\left[\Phi_r\right]\right]$, where $\sigma_c$ is the imaginary part of $\sigma^*(\mathbf{x}, \omega)$ and $\Phi_c$ is the imaginary potential. Thus, $\Phi_c$ is proportional to $\sigma_c$. Additionally, $\sigma_c$ decreased as $\omega$ increased; hence, the absolute value of the imaginary potential distribution (as shown in Figure 7c) decreased as $\omega$ increased.

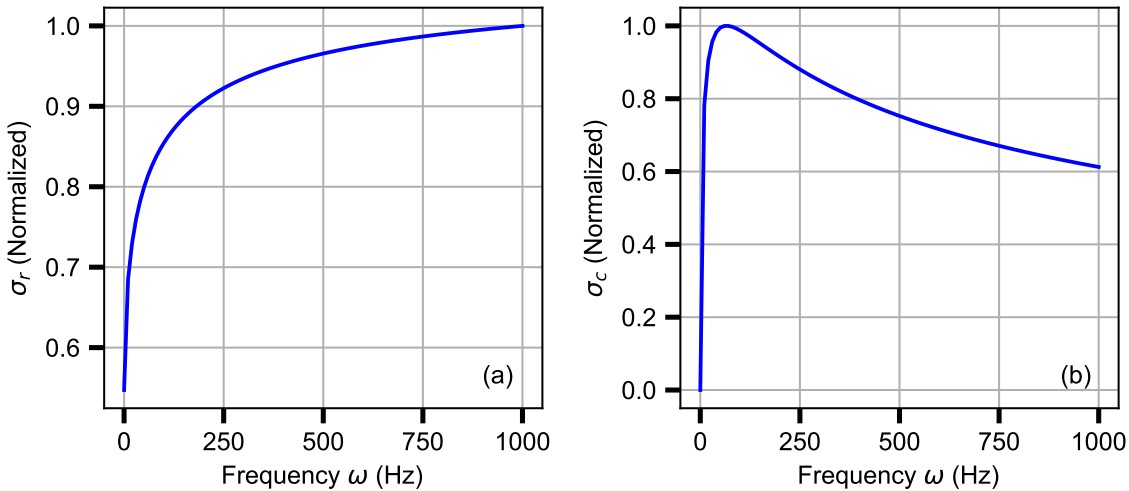

**Figure 8.** (**a**) Real, $\sigma_r$, and (**b**) imaginary, $\sigma_c$, components of complex conductivities vs. frequency. Each component of the complex conductivity was normalized with respect to its maximum value to better show trends.

Figure 7d shows the phase-shift data distribution along the same line as the tracer distribution, real and imaginary potential distribution. Mathematically, the phase shift is the inverse tangent of the ratio between the imaginary and real potential responses. Physically, it is the shift between the measured voltage and applied current signals that is largely governed by the polarization characteristics of the subsurface. In this study, phase shift leveraged signals from both real and imaginary potential responses in order to improve the interpretation of complex electrical impedance. From Figure 7, there was a change in phase shift where tracer transport was predominant. Moreover, the 1000 Hz frequency bounded the tracer zone better than lower frequencies that cannot be distinguished with ERT. This phase shift helped to constrain the polarized region or bound the interface between tracer-laden and tracer-free fluids. After identifying the region of interest according to these constraints, further geoelectrical interrogation could be performed with this volume. It is critical to mention that accurate estimation of the region of interest was found without performing a computationally expensive numerical inversion. Hence, through phase-shift signatures across multiple frequencies, the `PFLOTRAN-SIP` framework facilitates the identification of polarized or geochemically altered zones. The results have demonstrated that, if the subsurface contains polarized materials, then SIP will better capture signals than its counterpart ERT. For example, polarized materials, such as sulphide or oxide minerals, undergo electro-chemical reactions in response to additional current supplied by AC or SIP surveys [67]. This additional reaction could improve electrical signals and yield better results than ERT.

IP arises from solute transport and accumulation of ions/electrons in polarized materials (e.g., those with different grain types, colloids, biological materials, phase-separated polymers, blends, and crystalline minerals, etc.) when subject to an external electric field. Five mechanisms govern IP phenomena at frequencies <1 MHz: (1) Maxwell–Wagner polarization, which occurs at high frequencies [68–71]; (2) polarization of the inner part of the interface between minerals and water [72–74]; (3) polarization of the outer part of the interface between minerals and water [72,75]; (4) membrane polarization for multi-phase systems [65,76,77]; and, (5) electrode polarization observed in the presence of disseminated conductive minerals, such as sulfide minerals and pyrite [78–80].

*4.1. Limitations and Challenges*

The proposed framework has challenges and limitations that are similar to SIP geophysical techniques. Specifically, our `PFLOTRAN-SIP` simulations were geared toward IP mechanisms (1),

(4), and (5). Note that the Cole–Cole model that is given by Equation (11) neglects the effects of polarization at interfaces or sorption onto mineral surfaces. In order to simulate mechanisms (2) and (3), Equation (11) must be replaced with conductivity models that account for interface polarization with consideration of effective pore size, electrical formation factor, distribution of relaxation times, and sorption mechanisms [14,81]. This is one of limitations of the current framework. However, this challenge can be overcome by considering constitutive models that simulate interface polarization mechanisms. We note that our `PFLOTRAN-SIP` framework can easily account for such modifications in frequency-dependent electrical conductivity, which is beyond the scope of this work.

## 5. Conclusions

This work demonstrated the `PFLOTRAN-SIP` framework, which simultaneously simulates fluid flow, reactive transport, and SIP. A reservoir-scale tracer transport model demonstrated the proposed `PFLOTRAN-SIP` framework, where fluid flow and tracer concentration evolution were simulated over one year. Subsequently, we simulated 1062 electrical impedance at four frequencies. These simulations showed that contrasts in real potential were minimal, wven as the frequency varied. However, there was a significant change in the contrast of complex potentials across frequencies. Phase shift (a combination of real and complex potentials) helped to identify the region where tracer concentration was high. This analysis showed that SIP has two major advantages over ERT. First, SIP provides frequency-dependent electrical impedance data. Second, phase-shift signatures that were obtained from SIP analysis identified and constrained geochemically altered zones. Combining frequency-dependent real potential, complex potential, and phase responses from an SIP survey/simulation paints a more detailed picture of the subsurface with an enhanced ability to detect contaminants/tracers. Moreover, coupling fluid flow, reactive transport, and SIP models can better detect contaminants when compared to either the ERT or SIP method alone. For instance, through our numerical example, solute transport simulations provided insight into the tracer distribution. This information was used in order to customize SIP inversion to estimate frequency-dependent electrical conductivities, which yielded an improved image of tracer concentrations at different frequencies. Although this work focused on simulating tracer transport, it could also be applied to detect hydrocarbon flow, changes in the subsurface due to geochemical reactions, sulfide minerals, metallic objects, municipal wastes, and salinity intrusion. Moreover, this code could be used in feasibility studies for developing waste sequestration sites.

**Author Contributions:** B.A. developed the framework, ran models, and drafted the original manuscript. M.K.M. formulated the idea, supervised, and helped draft the manuscript. S.K. wrote the code and analyzed data. S.C.J. supervised, participated in drafting, and critically revised the manuscript. H.V. critically revised the manuscript. J.A.D. supervised and analyzed data. All authors have read and agreed to the published version of the manuscript.

**Funding:** This research was funded by the U.S. Department of Energy (DOE) Basic Energy Sciences (BES) and Fossil Energy (FE) programs. M.K.M., S.K., and B.A. also thank the support from Center for Space and Earth Sciences (CSES) Emerging Ideas R&D Program. The authors thank Glenn Hammond (Pacific Northwest National Laboratory) and Tim Johnson (Pacific Northwest National Laboratory) for the coupled framework `PFLOTRAN-E4D` upon which `PFLOTRAN-SIP` was built. During the paper writing process, M.K.M. is partially supported by the U.S. DOE's Office of Biological and Environmental Research (BER), Subsurface Biogeochemistry Research (SBR) program's Scientific Focus Area (SFA) at Pacific Northwest National Laboratory. Los Alamos National Laboratory is operated by Triad National Security, LLC, for the National Nuclear Security Administration of U.S. Department of Energy (Contract No. 89233218CNA000001).

**Conflicts of Interest:** The authors declare no conflicts of interests.

**Computer Code Availability, Installation, and Contribution:** The `PFLOTRAN-SIP` simulation input files used for this manuscript are available at the public github repository https://github.com/bulbulahmmed/PFLOTRAN-SIP. It also contains detailed instructions on installation the module. Additional information regarding the simulation datasets can be obtained from Bulbul Ahmmed (Email: `bulbul_ahmmed@baylor.edu`) and Maruti Kumar Mudunuru (Email: `maruti@pnnl.gov`).

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
