# Peer review of "PFLOTRAN-SIP: A PFLOTRAN Module for Simulating Spectral-Induced Polarization of Electrical Impedance Data"

_energies, doi:10.3390/en13246552_

Round 1

Reviewer 1 Report

I am qualified to comment on validation of such modeling.

In lines 197-221 in the section on PFLOTRAN Model Setup it appears the authors are putting together a scenario for an existing location. It would be useful in the experimental section to have done a validation/verification experiment on the migration of the contamination front. This could have been done without revealing the specific location or other details that may be classified.

The simple measurement of a volatile or semivolatile constituent in soil gas that would migrate vertically to the surface could provide the validation. This would be a two-dimensional projection to the surface, reflecting the contaminant plume below.

Although, this manuscript can likely be published as is, I feel the experiment is not yet finished. Modeling of processes that might be operational on other planets must leave the experimental component largely untested. However, on Earth, readily accessible site allow the opportunity for validation by direct measurements.

Author Response

The motivation of this numerical example was not to validate the proposed model and such a validation exercise is beyond the scope of this paper. The primary focus of this exercise was to describe and demonstrate how the proposed PFLOTRAN-SIP framework may be used to couple subsurface flow, tracer transport, and SIP processes.  We added text in introduction to highlight this point.
``The primary focus of this exercise was to demonstrate the capabilities of the PFLOTRAN-SIP framework. This study did not validate the synthetic numerical model because that is beyond the scope of this paper.'' Please see lines 78--80 in the revised manuscript.

Reviewer 2 Report

In my opinion, the article is well prepared. The only downside is the complete lack of pictures, which would be advisable for better understanding.

Author Response

Comment: In my opinion, the article is well prepared. The only downside is the complete lack of pictures, which would be advisable for better understanding.

Response: The authors thank the reviewer for recommending the manuscript.
It is unfortunate that the reviewer didn't get a chance to look at the figures.
The submitted manuscript has a total of nine figures.
Many of these figures have sub-figures illustrating the proposed approach.
This version of the manuscript contains figures.

Reviewer 3 Report

The submitted manuscript deals with PFLOTRAN based on spectral induced polarization ( PFLOTRAN-SIP) framework. The proposed PFLOTRAN-SIP takes into account various parameters simultaneously: fluid flow, reactive transport, and SIP. The paper is well-written, well-organized, and it is easy to follow. Additionally, the reported results considered various scenarios in both the time- and frequency- domains.

I have the following concerns:

  1. Abstract: The main goal of the abstract is to briefly highlight the research background, remaining unsolved challenge, proposed methodology, and the key findings. However, the current abstract is very long with so many details, please revised.
  2. Introduction: While the authors have performed throughout literature and highlighted the shortcoming existing in the current works, very recent references are missing. Firmly, the introduction objective is to ground the research question based on up-to-date literature. 
  3. Towards the end of the introduction section, please add two paragraphs. The first paragraph is to highlight the contribution of the paper. The second paragraph is to highlight how the rest of the manuscript is organized.
  4. Methodology: I will suggest the authors rename the third section “NUMERICAL MODEL SETUP” to “Methodology”. Then, a flow chart of the proposed framework can be added here to further improve the readability of the manuscript. Additionally, please bring Algorithm 1 from the Appendix to the newly named section, Methodology.
  5. RESULTS & DISCUSSION: The finding Figures are reported in the Appendix section, after the references, which makes it difficult for the reader to smoothly follow the analysis of the report results. Please consider bringing back these figures to the “RESULTS & DISCUSSION” section, and add a Table comparing the performance of the proposed technique with an available conventional method/ performance index.
  6. The authors have shown the efficacy of the proposed technique. What are the shortcomings? 
  7. Please comply with Energies Format Style (e.g. check “References” section).

Author Response

Comment #1: The submitted manuscript deals with PFLOTRAN based on spectral induced polarization

( PFLOTRAN-SIP) framework. The proposed PFLOTRAN-SIP takes into account various parameters

simultaneously: fluid flow, reactive transport, and SIP. The paper is well-written, well-organized, and it

is easy to follow. Additionally, the reported results considered various scenarios in both the time- and

frequency- domains. I have the following concerns:

Abstract: The main goal of the abstract is to briefly highlight the research background, remaining

unsolved challenge, proposed methodology, and the key findings. However, the current abstract is very long with so many details, please revise.

Response: The authors thank the reviewer for recommending the manuscript. We have now modified

our abstract to incorporate this suggestion. Modified abstract is as follows: “Spectral induced polarization

(SIP) is a non-intrusive geophysical method that collects chargeability information (the ability of a

material to retain charge) in the time domain or its phase shift in the frequency domain. Although SIP

is a temporal method, it cannot measure the dynamics of flow and solute/species transport in the subsurface over long times (i.e,. 10–100s of years). To capture long-term dynamics, data collected with the SIP technique need to be coupled with fluid flow and reactive-transport models. To address this challenge, PFLOTRAN-SIP was built to couple SIP data to fluid flow and solute transport processes. Specifically, this framework couples the subsurface flow and transport simulator PFLOTRAN and geoelectrical simulator E4D without sacrificing computational performance. PFLOTRAN solves the coupled flow and solute-transport process models to estimate solute concentrations, which were used in Archie’s model to compute bulk electrical conductivities at near-zero frequency. These bulk electrical conductivities were modified using the Cole-Cole model to account for frequency dependence. Using the estimated frequency-dependent bulk conductivities, E4D simulated the real and complex electrical potential signals for selected frequencies for SIP. These frequency-dependent bulk conductivities contain information relevant to geochemical changes in the system. This study demonstrated that the PFLOTRAN-SIP framework is able to detect the presence of a tracer in the subsurface. SIP offers a significant benefit over ERT in the form of greater information content. It provided multiple datasets at different frequencies that better constrained the tracer distribution in the subsurface. As a result, this framework allows practitioners of environmental hydrogeophysics and biogeophysics to monitor the subsurface with improved resolution.”

Comment #2: Introduction: While the authors have performed throughout literature and highlighted

the shortcomings existing in the current works, very recent references are missing. Firmly, the introduction

objective is to ground the research question based on up-to-date literature.

Response: Agreed. We added a few recent results in this area including the following text: “tree

root finding [1], prospecting alpine permafrost [2], investigating seawater intrusion [3], finding preferential infiltration in loess [4], and monitoring internal erosion processes [5].” Please see lines 44–46. Also we added the following text that shows the recent code development for SIP but not for a coupled system: “IP-decay model [6]” Please see line 64.

Comment #3: Towards the end of the introduction section, please add two paragraphs. The first

paragraph is to highlight the contribution of the paper. The second paragraph is to highlight how the rest

of the manuscript is organized.

Response: Agreed. We have now added text in the last of the introduction section. Please see

lines 74–97: The main contribution of this study was development of a framework to include SIP in a

module called PFLOTRAN-SIP. This module couples subsurface flow and transport processes in PFLOTRAN with SIP capabilities in E4D. This framework adds to the recently developed PFLOTRAN-E4D code and leverages the PFLOTRAN-E4D massively parallel capabilities. We demonstrated PFLOTRAN-SIP capabilities with a representative tracer-transport process in a medium with polarization properties. The primary focus of this modeling exercise was to illustrate the capabilities of the PFLOTRAN-SIP framework not to validate the proposed model.

The paper is organized as follows: Sec. 1 discusses the limitations of ERT along with the advantages

of the SIP method. The importance of coupling between fluid flow, reactive-transport, and SIP process

models is discussed. Discussion on the state-of-the-art simulators is also provided. Sec. 2 introduces the

PFLOTRAN-SIP framework. The methodology for coupling PFLOTRAN and E4D simulators is described. Process models related to SIP, fluid flow, and solute transport in PFLOTRAN and E4D simulators are presented. Also, this section discusses the mesh interpolation that transfers state variables between the PFLOTRAN and E4D meshes. Sec. 3 describes a reservoir-scale model setup to perform high-fidelity numerical simulations. It describes the model domain, related meshes, initial conditions, boundary conditions, and various input parameters related to the PFLOTRAN-SIP framework. Additionally, this section provides details on the inversion procedure for electrical conductivity at different frequencies. This exercise was performed with the intention of comparing the simulated conductivity/tracer distribution from the PFLOTRAN-SIP framework and inverted electrical conductivity from SIP inversion. Sec. 4 explains simulated electrical potentials for a measurement and compares the true/simulated and estimated conductivities. It shows that estimated frequency-dependent conductivities from the SIP inversion were consistent with the tracer concentration/simulated conductivity from the PFLOTRAN-SIP framework. Sec. 4 also provides a discussion on the numerical results, limitations of the proposed framework, and how geoscientists can use the PFLOTRAN-SIP framework for their applications. Finally, conclusions are drawn in Sec. 5.

Comment #4: Methodology: I will suggest the authors rename the third section “NUMERICAL

MODEL SETUP” to “Methodology”. Then, a flow chart of the proposed framework can be added here

to further improve the readability of the manuscript. Additionally, please bring Algorithm 1 from the

Appendix to the newly named section, Methodology.

Response: Agreed. We have implemented the suggestions. Please see updated Sec. 3 in the revised

manuscript.

Comment #5: RESULTS & DISCUSSION: The finding Figures are reported in the Appendix section,

after the references, which makes it difficult for the reader to smoothly follow the analysis of the report results. Please consider bringing back these figures to the RESULTS & DISCUSSION section, and add a Table comparing the performance of the proposed technique with an available conventional method/performance index.

Response: The authors thank the reviewer for the suggestion. We have brought our figures and algorithm

into the results and discussion section. Please see the updated Sec. 4 in the revised manuscript.

Comment #6: The authors have shown the efficacy of the proposed technique. What are the shortcomings?

Response: We have mentioned our limitations in the revised manuscript. Please see lines 388–397.

“The proposed framework has challenges and limitations that are similar to SIP geophysical techniques.

Specifically, our PFLOTRAN-SIP simulations were geared toward IP mechanisms (1), (4), and (5). Note

that the Cole-Cole model given by Eq. (11) neglects the effects of polarization at interfaces or sorption

onto mineral surfaces. To simulate mechanisms (2) and (3), Eq. (11) must be replaced with conductivity

models that account for interface polarization with consideration of effective pore size, electrical formation factor, distribution of relaxation times, and sorption mechanisms [7, 8]. This is one of limitations of the current framework. However, this challenge can be overcome by considering constitutive models that simulate interface polarization mechanisms. We note that our PFLOTRAN-SIP framework can easily account for such modifications in frequency-dependent electrical conductivity, which is beyond the scope of this work.”

Comment #7: Please comply with Energies Format Style (e.g., check “References” section).

Response: Agreed. We have now amended our references to follow the Energies’ format.

Reviewer 4 Report

This study demonstrates coupling PFLOTRAN and SIP through the coupled PFLOTRAN-E4D and represents the capability of the coupled tools to predict the solute transport simulations and the tracer distribution. The manuscript provides a nice summary of the needs of this study.

The coupling has been introduced very well and the coupled tool applied for a reservoir-scale tracer transport model to study the fluid flow and tracer concentration evolution over time. The simulation results showed the advantages of the SIP compared to ERT to capture geochemically altered zones after coupling the reactive transport code by providing frequency-dependent real, complex potential, and phase responses from the SIP survey.

The manuscript is well written. I only have a few minor suggestions.

  • Section 3.1: That would be great if the parameters and values are presented in the Table.
  • Section “Computer Code Availability, Installation, and Contribution”: That would be good if the author only refers to the PFLOTRAN-SIP GitHub and avoids presenting the code installation and … in the manuscript.
  • It would be better if the author could further explain how the coupled tool can detect the mineral geochemical reaction (e.g. sulfide)
  • The author mentioned that it seems that “there is no simulator in the open-source literature that couples fluid flow, solute transport, and SIP process models to analyze geoelectrical signatures in large-scale systems”, however, is there any solution to validate the results.

Author Response

General Comment: This study demonstrates coupling PFLOTRAN and SIP through the coupled

PFLOTRAN-E4D and represents the capability of the coupled tools to predict the solute transport simulations and the tracer distribution. The manuscript provides a nice summary of the needs of this study. The coupling has been introduced very well and the coupled tool applied for a reservoir-scale tracer transport model to study the fluid flow and tracer concentration evolution over time. The simulation results showed the advantages of the SIP compared to ERT to capture geochemically altered zones after coupling the reactive transport code by providing frequency-dependent real, complex potential, and phase responses from the SIP survey. The manuscript is well written. I only have a few minor suggestions.

Response: Thanks for recommending our work. We have addressed your suggestions in the revised

paper.

Minor Comment #1: Section 3.1: That would be great if the parameters and values are presented in the Table.

Response: Agreed. We added Table 1 to the revised manuscript; it lists parameters and values used in the numerical model. Please see the response to reviewer’s comment file and the updated manuscript.

Minor Comment #2: Section “Computer Code Availability, Installation, and Contribution”: That would be good if the author only refers to the PFLOTRAN-SIP GitHub and avoids presenting the code installation and . . . in the manuscript.

Response: Agreed. We removed the instructions in this section and moved them to our GitHub repository: https://github.com/bulbulahmmed/PFLOTRAN-SIP.

Minor Comment #3: It would be better if the author could further explain how the coupled tool can detect the mineral geochemical reaction (e.g. sulfide)

Response: We added the following text to the revised manuscript to clarify this point. Please see lines 376–379 in the revised manuscript. “Results have demonstrated that if the subsurface contains polarized materials, then SIP will better capture signals than its counterpart ERT. For example, polarized materials such as sulphide or oxide minerals undergo electro-chemical reactions in response to additional current supplied by AC or SIP surveys [9]. This additional reaction could improve electrical signals and yield better results than ERT.”

Minor Comment #4: The author mentioned that it seems that “there is no simulator in the opensource literature that couples fluid flow, solute transport, and SIP process models to analyze geoelectrical signatures in large-scale systems,” however, is there any solution to validate the results.

Response: There are solution procedure to validate the results. Given experimental and field data on flow, transport, and voltage measurements, we can perform model calibration and validation exercises. However, such efforts are beyond the scope of the current paper. This is now specifically mentioned in Section 1. Please see lines 74–80 in the revised manuscript.

Round 2

Reviewer 3 Report

The authors have answered all questions and done all required modifications. I suggest that the current version of the paper is adequate to be published in Energies.